# A Comparative Sentiment Analysis of Greek Clinical Conversations Using BERT, RoBERTa, GPT-2, and XLNet

**DOI:** 10.3390/bioengineering11060521

**Published:** 2024-05-21

**Authors:** Maria Evangelia Chatzimina, Helen A. Papadaki, Charalampos Pontikoglou, Manolis Tsiknakis

**Affiliations:** 1Department of Electrical and Computer Engineering, Hellenic Mediterranean University & Institute of Computer Science, Foundation for Research and Technology–Hellas (FORTH), 70013 Heraklion, Greece; tsiknaki@ics.forth.gr; 2Department of Hematology, School of Medicine, University of Crete, 71003 Heraklion, Greece; e.papadaki@uoc.gr (H.A.P.); xpontik@uoc.gr (C.P.)

**Keywords:** sentiment analysis, healthcare, clinical dialogues, cancer, hematologic malignancies, Greek, palliative care, deep learning, natural language processing

## Abstract

In addressing the critical role of emotional context in patient–clinician conversations, this study conducted a comprehensive sentiment analysis using BERT, RoBERTa, GPT-2, and XLNet. Our dataset includes 185 h of Greek conversations focused on hematologic malignancies. The methodology involved data collection, data annotation, model training, and performance evaluation using metrics such as accuracy, precision, recall, F1-score, and specificity. BERT outperformed the other methods across all sentiment categories, demonstrating its effectiveness in capturing the emotional context in clinical interactions. RoBERTa showed a strong performance, particularly in identifying neutral sentiments. GPT-2 showed promising results in neutral sentiments but exhibited a lower precision and recall for negatives. XLNet showed a moderate performance, with variations across categories. Overall, our findings highlight the complexities of sentiment analysis in clinical contexts, especially in underrepresented languages like Greek. These insights highlight the potential of advanced deep-learning models in enhancing communication and patient care in healthcare settings. The integration of sentiment analysis in healthcare could provide insights into the emotional states of patients, resulting in more effective and empathetic patient support. Our study aims to address the gap and limitations of sentiment analysis in a Greek clinical context, an area where resources are scarce and its application remains underexplored.

## 1. Introduction

The application of natural language processing (NLP) in healthcare, particularly sentiment analysis, offers a transformative approach to patient-centered care, particularly for patients affected by hematologic malignancies. The application of sentiment analysis in this domain provides a better understanding of patient experiences but also offers new opportunities for enhancing patient–clinician communication. Such advancements are crucial in environments with diverse linguistics where the emotional context of health conversations can be particularly complex. Moreover, integrating sentiment recognition in healthcare systems plays a crucial role in affective computing, thereby contributing to the development of more empathetic human–computer interaction systems.

Our study aims to unravel the complex emotional narratives that are crucial yet often overlooked in clinical settings. Sentiment analysis is crucial for decoding emotions and sentiments from text and is applied in diverse fields such as politics, marketing, public policy, disaster management, and public health. However, the integration of sentiment analysis in diverse linguistically settings, such as Greek healthcare, remains underexplored. The objective of our research is to fill that gap by applying advanced language models to understand patient emotions in Greek patient–clinician conversations related to hematologic malignancies.

Decoding the emotional context which plays vital role in patient care could offer insights into how healthcare professionals can tailor their communication strategies. Hematologic malignancies arise from the uncontrolled proliferation of abnormal blood cells, leading to a disruption in the production and function of normal blood cells. Variants of hematologic malignancies encompass myeloproliferative neoplasms (MPNs), myelodysplastic syndromes (MDSs), acute myeloid leukemias (AML), B- and T-cell lymphoblastic leukemias/lymphomas, and Hodgkin’s lymphoma [1]. These conditions not only challenge patients medically but also weigh heavily on their emotional and psychological well-being [2,3]. By investigating the sentiments expressed during these conversations, we aim to shed light on fostering more empathetic approaches to patient care. This exploration not only advances our understanding of sentiment analysis in healthcare but also underscores the importance of cultural and linguistic considerations in patient care.

Building on the foundational work of our previous study, this paper extends the exploration of sentiment analysis within the context of Greek patient–clinician dialogues, specifically focusing on hematologic malignancies [4]. With an expanded dataset comprising over 43,000 utterances between Greek patients and clinicians, focusing on hematologic malignancies, we employ state-of-the-art NLP models like BERT [5], RoBERTa [6], GPT-2 [7], and XLNet [8] to capture the emotional context in the Greek language. Moreover, this study addresses the challenges of conducting sentiment analysis in languages like Greek that have limited digital resources and complex nuances. Comparing these models will help to identify the most effective sentiment analysis methods for this unique language context, contributing valuable insights to the field of computational linguistics in healthcare.

In addition to improving the technical aspects of sentiment analysis, our study explores the implications of our findings for AI-powered conversational agents in healthcare contexts. By identifying and responding to patients’ emotional needs, these tools could improve patient care by offering support that is both empathetic and tailored to individual needs. Our goal is to advance the understanding of emotional contexts in order to develop more empathetic healthcare tools and underline the potential transformation NLP techniques could provide in patient care strategies.

Despite the growing interest in applying NLP technologies in healthcare, Greek clinical dialogues remain significantly underrepresented in sentiment analysis research. This study introduces the first dataset of clinical dialogues in the Greek language and fills a critical gap in the literature. Our findings indicate that BERT outperforms other models in emotional nuances in Greek patient–clinician dialogues. This underscores BERT’s potential in improving sentiment analysis in healthcare communications significantly, enabling clinicians to better understand and respond to their patients’ emotional states. The success of BERT highlights the opportunity to develop more refined and empathetic AI-driven tools such as conversational agents in healthcare settings, especially beneficial for Greek-speaking populations.

## 2. Related Work

Sentiment analysis involves recognizing the emotion expressed in textual data, holding significant relevance across various fields, including healthcare. Several methodologies for effectively extracting sentiment information have been explored by researchers.

These sentiment analysis methods can be supervised, unsupervised, or both. The supervised approach employs machine-learning models trained using labeled datasets, where text is labeled according to the emotion expressed. This allows the models to effectively predict new unlabeled data. Examples include traditional machine-learning-based methods [9,10,11] and deep-learning-based methods [12,13]. On the other hand, the unsupervised approach utilizes techniques such as clustering, topic modeling, or lexicon approaches which do not require labeled datasets [14,15].

In recent years, sentiment analysis in the healthcare domain has garnered increasing academic interest. Numerous studies have investigated health-related issues such as cancer, mental health, chronic conditions, eating disorders, addiction, pain, infectious diseases, and quality of life [16]. These studies predominantly rely on data collected from mainstream social media platforms like Twitter and employ machine-learning approaches [17]. Interest in sentiment analysis for the Greek language has increased due to the growing volume of Greek textual content on several social networking platforms. Academic studies across diverse fields, including informatics, management, business administration, political sciences, computer engineering, and statistics, have explored sentiment analysis in Greek [18,19,20,21,22]. However, a small number of studies have focused on the health domain and are limited only to COVID-19 topics [23,24,25,26,27]. All studies utilized data by mining the Greek web or social media platforms such as Twitter.

In this study, we attempt to fill the existing gap in sentiment analysis methodologies applied to Greek clinical conversations by creating a corpus originating from authentic dialogue. In our research, we investigate state-of-the-art deep-learning models, including BERT, RoBERTa, XLNet, and GPT-2, utilizing a comprehensive dataset. Building upon previous research, which showed the superiority of BERT in sentiment analysis tasks, our study extends the comparison to include additional deep-learning architectures. Through this comparative analysis, we aim to provide valuable insights into the effectiveness of these advanced models in capturing sentiment nuances within clinical conversations.

## 3. Materials and Methods

In this research, we compared state-of-the-art natural language processing (NLP) models including BERT, RoBERTa, XLNet, and GPT-2 based on their proven effectiveness in sentiment analysis. The first choice was BERT, as it showed promising results in our previous research. The BERT-base-uncased model consists of 12 transformer layers. Each layer has 12 attention heads and 768 hidden units, totaling around 110 million parameters. Next choice was Robustly Optimized BERT Approach (RoBERTa) which is an extension of BERT with improved training methodology trained on larger sequences and larger corpus. The RoBERTa-base model has 125 million parameters overall and is composed of 12 transformer layers with 768 hidden units and 12 attention heads. RoBERta was chosen due to its improved training methodology compared to BERT, which could have potential to improve the accuracy of sentiment analysis. XLNet is also a transform-based language model, but it introduces a permutation-based training approach that examines words in different orders and was trained with a larger dataset. The XLNet-base-cased model consists of 12 transformer layers and 12 attention heads with 768 hidden units resulting in 110 million parameters. This approach allows XLNet to understand text from both directions more efficiently, therefore capturing complex patterns in text data. This feature could potentially enhance the results of sentiment analysis in complex scenarios such as clinical conversations. Finally, GPT-2 is another transformation-based model that processes text from left to right and generates related text based on previous words. The configuration of GPT-2 includes 12 layers, with 768 hidden units and 12 attention heads, summing up to about 117 million parameters. The GPT-2 was included in the study for its generative capabilities and ability to capture contextual nuances that could provide unique insights on emotion recognition within clinical context.

### 3.1. Data Collection

To conduct sentiment analysis, it is important to have access to real medical conversations related to hematologic malignancies. We collected dialogues between healthcare professionals and patients diagnosed with hematologic malignancies at the University Hospital of Heraklion using a voice recorder. Ethical approval for the study was obtained from the University Hospital’s Ethical Committee. Prior to participation, all patients were briefed on the study’s objectives and provided their consent. The dataset includes initial consultations between patients and clinicians, as well as subsequent follow-up sessions occurring every 3, 6, or 12 months. Initial consultations primarily focus on gathering medical and family history, while follow-up visits collect symptom monitoring and assessments of quality of life.

The total duration of the recorded dialogues amounts to 185 h, resulting in a dataset comprising 43,207 utterances, 48,393 sentences, and 527,622 words. The term “utterance” is used to describe a discrete verbal contribution or statement made by either the healthcare professional or the patient. Each utterance may contain a single sentence or multiple sentences, reflecting the detail or complexity of the information shared. Essentially, an utterance marks a participant’s individual contribution or turn in the conversation. When conducting sentiment analysis or other types of textual analysis, each utterance is treated as an independent unit for examination, with its specific content and sentiment analyzed separately from other utterances.

### 3.2. Data Preprocessing

The collected audio recordings were manually trascribed since speech-to-text (STT) techniques [28] failed due to noisy environment and dynamic conversation turns. Moreover, all personally identifiable information was removed in order to maintain patients’ anonymity. In the trascribed data, line changes and tags were employed in order to identify speaker turns. Finally, the annotated sentiment was added as tag at each utterance.

The transcribed textual data were transcribed in three categories: neutral, negative, and positive. A detailed set of guidelines was followed during the annotation process and the utterances were categorized based on the expressed sentiment. To maintain the consistency of sentiment labeling, two annotators underwent training according to these guidelines and were provided with a comprehensive manual detailing the criteria for assigning sentiments into one of the three categories. In order to achieve a consistent sentiment labeling, crucial step was the identification of the predominant sentiment in each utterance. Moreover, the annotators carefully evaluated the utterances, to ensure the annotation was objective and did not reflect their own personal agreement or disagreement with the speaker’s point of view. When annotators did not agree on an emotion label, they participated in a meeting to discuss their views and align on the correct classification based on the guidelines.

Utterances were categorized as positive when positive sentiments where expressed, such as satisfaction, relief, happiness, etc. Moreover, positive sentiments include utterances where the speaker expressed admiration or gratitude towards the clinician. Negative sentiments include emotions such are anger, pain, anxiety, etc. Finally, if an utterance lacks emotional expression or provides factual information, it is categorized as neutral sentiment. In neutral category were also included general inquiries or educational information expressed by the clinicians to inform patients regarding their health condition. The distribution of sentiment categories is shown in Figure 1, revealing that 35.85% are categorized as positive sentiments, 13.17% as negative sentiments, and 50.98% as neutral sentiments.

Datasets frequently are imbalanced, where one sentiment, such as neutral, outweights the other two sentiments. This imbalance can vary significantly which results in challenges in sentiment analysis. Our dataset is imbalanced since it captures genuine communication between clinicians and patients. Crucial aspect of our research is to maintain the authenticity of the dialogues; therefore, we refrained from artificially balancing the dataset, recognizing that this process could introduce biases and distortions.

### 3.3. Tokenization

Tokenization is an important preprocessing step in the training of models for processing natural language. It involves breaking down the text into smaller parts known as tokens, such as words, characters, or other meaningful elements. In our study, we used the tokenizers designed for each model separately.

For BERT, the tokenization process is based on WordPiece tokenizer which splits words into subwords known as “wordpieces”. RoBERTa follows the same architecture as BERT but uses a byte-level BPE tokenizer, which merges the most frequent pairs of characters as a single token in the model’s vocabulary. GPT-2 also uses a byte pair encoding (BPE) tokenizer but it is a generative model that predicts the next word of a sequence. Finally, XLNet uses a unique permutation-based tokenization method that rearranges the words in a sentence and uses the last words to predict the next word in the sequence.

### 3.4. Model Training

In order to achieve an unbiased comparison across all models, we followed a standardized training procedure. For consistency, all models in this study were fine-tuned using the same training parameters. These parameters included using the AdamW optimizer, an early stopping mechanism, and the same batch size, number of training epochs, and data splits.

#### 3.4.1. Data Preparation

The dataset was split into 80% for training, 10% for validation, and 10% for testing before model training. This step is crucial for ensuring an unbiased comparison of each model’s performance, since each model is evaluated using the same subsets. Since the corpus was specifically created for this study, we have already excluded special characters. Finally, the text was not converted to lowercase since we wanted to maintain the original casing which is important in Greek language.

#### 3.4.2. Handling of Unbalanced Data

The class imbalance of our dataset represents genuine interactions in clinical settings. To address the imbalanced dataset, we employed class weighting and utilized appropriate evaluation metrics. This approach ensures that the models effectively represent the true distribution of sentiments in the clinical conversations. The weighting process includes weight calculations based on the training data distribution in order to encourage the models to learn equally from all sentiment categories and avoid biased model predictions. For example, the negative sentiment which is the minority class in our dataset, is assigned higher weights in order to help the model generalize effectively.

#### 3.4.3. Early Stopping Mechanism

In order to avoid overfitting, an early stopping mechanism was used during the model training phase. This mechanism monitors the performance of the model on the validation set and stops the training process if no improvement is observed after 3 epochs. This practice of early stopping prevented the models from training unnecessarily and losing their ability to generalize when processing unseen data.

#### 3.4.4. Fine Tuning

Each model was consistently trained using the same sets for training, validation, and testing. We used the AdamW optimizer to efficiently tune each model’s parameters with a learning rate that was determined empirically through testing various values [29]. In order to avoid overfitting, the models were trained for 10 epochs with an early stopping mechanism. To evaluate our models, we applied a 10-fold cross-validation technique which divides the dataset into ten parts, or “folds”, using nine for training and one for validation. This process is repeated ten times, so that each time a different fold is used for validation. This approach provides a more comprehensive evaluation of the models’ performance and ensures that the evaluation is not biased by any specific subset. By using 10-fold cross-validation to evaluate the model, we gain valuable insights into its generalization abilities and robustness.

The training process was performed using the PyTorch library, known for its efficiency and flexibility in training complicated neural networks [30]. For model architecture and natural language processing tasks, we used the Transformers library by Hugging Face, which provides access to pre-trained models and their tokenizers [31]. The training process used four NVIDIA GeForce RTX 2080 Ti GPUs with a batch size of 16 and it ranged from 5 to 7 h.

#### 3.4.5. Evaluation Metrics

The performance of each model was evaluated using standard metrics, including accuracy, precision, recall (or sensitivity), F1-score, and specificity in the test set. Each of these metrics provide unique insights regarding model’s performance and effectiveness in sentiment classification. The mathematical expressions of the evaluation metrics are defined below:(1)$$Accuracy=\frac{TP+TN}{TP+TN+FP+FN}$$
(2)$$Sensitivity\;or\;recall=\frac{TP}{TP+FN}$$
(3)$$Specificity=\frac{TN}{TN+FP}$$
(4)$$Precision=\frac{TN}{TP+FP}$$
(5)$$F1-Score=2\times \frac{Precision\times Recall}{Precision+Recall}$$

The simplest of these metrics is accuracy, which measures the ratio of correctly predicted observations to the total observations. It is essentially a measure of the model’s overall ability in accurately classifying both positive and negative instances. In general, a model with higher accuracy is generally considered more reliable for classification tasks.

Precision provides information about the model’s accuracy in predicting positive labels, by comparing correct positive predictions by the total number of positive predictions (true positives and false positives). This metric is particularly crucial when the cost of a false positive is substantial.

Recall, or sensitivity, calculates the model’s capability to identify all true positives from the dataset. It achieves this by dividing the number of true positives by the sum of true positives and false negatives. High recall shows the model’s effectiveness in capturing positive instances.

The F1-score serves as a balanced mean between precision and recall, providing a better understanding of model’s performance. It is particularly useful when trying to achieve a balance between precision and recall, especially when dealing with imbalanced datasets.

Finally, specificity assesses the model’s capability in correctly identifying negative instances. This metric is calculated by dividing the number of true negatives by the sum of true negatives and false positives. This evaluation metric is important to ensure that negative instances are not labeled as positive, which is critical in sentiment analysis.

In order to have an unbiased comparison of the models, we followed a standardized training procedure by using consistent data splits and evaluation metrics. This approach ensured that any observed differences in performance could be attributed to the model’s features and capabilities rather than variations in training process.

## 4. Results

Our study performed a comparison of sentiment analysis in Greek clinical conversations using four different language models: BERT, RoBERTa, GPT-2, and XLNet. The accuracy, F1-score, precision, specificity, and recall were used to the evaluate the performance of each model in three sentiment categories: negative, neutral, and positive. In the following sections, we provide the evaluation metrics of each language model, starting with the BERT model. These results will highlight the strengths and limitations in the context of sentiment analysis in Greek clinical conversations.

### 4.1. Results of BERT Model

The model that achieved the highest results in all sentiment categories was BERT. In addition, BERT achieved a high specificity of 0.983 in negative sentiments which indicates its effectiveness in avoiding the misclassification of non-negative sentiments. The model achieved an accuracy of 0.9548, showing its effectiveness in capturing sentiment nuances in clinical conversations. The results are presented in Table 1.

### 4.2. Results of RoBERTa Model

RoBERTa exhibited a strong performance, particularly in capturing neutral sentiments. The model achieved an accuracy of 0.9143, demonstrating its reliability in sentiment classification tasks, especially in identifying neutral sentiments. Moreover, it achieved a specificity of 0.9765 in negative sentiments which demonstrates its capability to not mistake other sentiments for negative ones. This feature is crucial in clinical settings where incorrectly identifying a sentiment as negative could impact patient care. The results are shown in Table 2.

### 4.3. Results of GPT-2 Model

GPT-2 exhibited a moderate performance, with varying results for precision, recall, and F1-scores across sentiment categories. The model showed a higher accuracy in identifying neutral sentiments compared to negative and positive sentiments. The model struggles in identifying negative sentiments based on the evaluation results but its specificity shows the capability to avoid the misclassification of non-negative sentiments. The results are available in Table 3.

### 4.4. Results of XLNet Model

XLNet’s performance was moderate, with precision, recall, and F1-scores differing across sentiment categories. Moreover, the model showed moderate results in specificity which indicates a difficulty in avoiding false positives, especially in the negative category. The model displayed a higher accuracy in recognizing neutral sentiments compared to negative and positive sentiments. The results are shown in Table 4.

In our evaluation of language models for sentiment analysis in Greek clinician conversations, BERT and RoBERTa exhibited robust results in all three sentiment categories. BERT outperformed the other models across all sentiment categories (negative, neutral, and positive), indicating its effectiveness in accurately identifying and classifying sentiment. RoBERTa showed promising results capturing neutral sentiments, demonstrating precision and recall rates above 0.94. Regarding GPT-2 and XLNet, the results are showing average performance. GPT-2 showed a moderate accuracy, which is particularly stronger in identifying neutral sentiments but less precise in detecting negative sentiments. XLNet struggled with precision in negative and positive sentiments compared to neutral. We also considered specificity, which is crucial because it ensures that the models are not only capable of recognizing the correct sentiments (like positive, negative, or neutral) but are also effective in not mistakenly categorizing unrelated sentiments as part of these categories.

These findings underscore the importance of selecting appropriate language models tailored to the specific requirements of sentiment analysis in clinical conversations. BERT and RoBERTa showed a robust performance which suggests their suitability for tasks demanding sentiment understanding, while GPT-2 and XLNet, may require further optimization or context-specific tuning for optimal performance in clinical settings. Overall, our evaluation emphasizes the need for thoughtful model selection based on performance characteristics and task-specific considerations, ensuring reliable sentiment analysis outcomes in clinical contexts.

## 5. Discussion

Sentiment analysis in clinical text is an essential part of affective computing. We investigated four different models’ performance, BERT, RoBERTa, GPT-2, and XLNet, in the sentiment classification of our manually annotated dataset related to hematologic malignancies. Each model exhibited various results which reveals their strengths, disadvantages, and potential implications for clinical applications.

BERT outperformed the other models in all emotion categories, with particularly high scores for negative and positive emotions. The successful detection of negative and positive emotions is crucial in clinical settings as it can provide insight into patients’ emotional states. This shows that BERT is effective in correctly classifying emotions in clinical conversations focused on hematologic malignancies. RoBERTa showed promising results, especially in the neutral sentiment category, which suggests that it could be suitable for applications where neutral emotion detection is important. GPT-2 and XLNet showed more modest results, showing strengths in neutral emotion recognition but a lower accuracy in negative and positive emotion recognition. These results suggest the potential limitations of these models in handling the complex emotional range of clinical conversations about hematologic malignancies.

The moderate performance of GPT-2 and XLNet compared to BERT and RoBERTa, could be attributed to several factors. The design of GPT-2 focuses on generating text based on the given context and previous words, which may not be suitable for sentiment analysis. Similarly, XLNet, although bidirectional, employs a permutation language modeling approach that may not be as affective for sentiment analysis in specialized domains like hematologic oncology. Another aspect to consider is the size and specificity of the fine-tuning dataset. GPT-2 and XLNet might require larger, more domain-specific datasets to better adapt to the language and sentiment expressions typical of clinical dialogues. An indicator that a larger dataset might be required is that both models showed especially low results in the negative category, which had the smallest distribution of the dataset. Future research could explore tailored fine-tuning strategies and larger datasets to enhance GPT-2 and XLNet’s performance in sentiment analysis within specialized clinical contexts.

Moreover, another approach could be the combination of multiple models which could have the potential to improve overall performance, offering more robust sentiment analysis tools for healthcare professionals. Additionally, investigating techniques to refine these models using expanded and varied datasets tailored to hematologic malignancies could lead to more precise and clinically meaningful outcomes. Sentiment analysis holds significant potential in improving patient care by providing healthcare professionals with insights into patients’ emotions. The ability to detect negative emotions in real time can enable early interventions, while the recognition of positive emotions can indicate patient satisfaction. This analysis is crucial for identifying moments of emotional distress during clinical interactions, which is valuable for improving patient well-being. Developing an AI tool incorporating the best-performing sentiment analysis model could analyze conversations and provide feedback to healthcare professionals, potentially improving patient care. Additionally, sentiment analysis could be incorporated into medically focused conversational agents. These AI-based tools could be designed to help patients by offering empathetic responses and tailored information based on patients’ emotional states. Tools that enable sentiment analysis could improve patient-centered solutions, which are vital in palliative care.

Our comparative study successfully recognized sentiments in clinical conversations focused on hematologic malignancies. Since positive feelings are crucial for patient satisfaction, it is important to note that an excessive focus on positivity, known as toxic positivity, can occasionally be harmful. Patients with hematologic malignancies often encounter complex emotions and an overemphasis on positive feelings may potentially overlook real emotions of fear, pain, and uncertainty. Incorporating sentiment analysis tools that are sensitive to negative or mixed emotions could help clinicians identify underlying negative sentiments more effectively. Finally, while positive sentiment is a sign of effective communication, it is important to ensure that it does not overshadow the full range of expressed emotions.

Finally, it is important to acknowledge the limitations of our study in interpreting its results and implications. The data we used might not fully represent the diversity of clinical conversations in Greece, as it was collected from a single medical institution and specific groups of healthcare professionals and patients. This limitation could restrict the range of clinical interactions captured in our study, potentially impacting the applicability of our findings to broader healthcare settings.

## 6. Conclusions

Our future work will involve exploring the practical applications of sentiment analysis in clinical settings, focusing on real-time application during patient interactions by involving conversational agents. Moreover, we aim to expand our dataset by collaborating with other clinical centers. Our ultimate goal is to enhance patient care by identifying emotional distress, and personalizing communication approaches to better suit individual patient needs, through the deployment of conversational agents.

In summary, this study contributes valuable insights into the application of state-of-the-art transformer-based models in the sentiment analysis of clinical conversations in the Greek language. Leveraging these findings to develop conversational agents tailored to hematologic malignancies could significantly enhance patient communication and care delivery in healthcare settings.

## Figures and Tables

**Figure 1 bioengineering-11-00521-f001:**
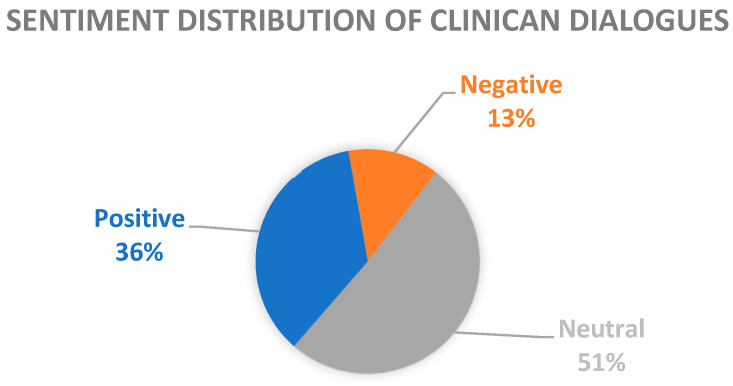
Sentiment distribution of clinical dialogues.

**Table 1 bioengineering-11-00521-t001:** BERT results.

BERT	Results			
	Precision	Recall	F1-Score	Specificity
Negative	0.9005	0.9006	0.8999	0.9863
Neutral	0.9736	0.9637	0.9686	0.9701
Positive	0.9459	0.9609	0.9532	0.9708
Accuracy	0.9548			
Precision	0.9550			
Recall	0.9548			
F1-Score	0.9547			

**Table 2 bioengineering-11-00521-t002:** RoBERTa results.

Bert	Results			
	Precision	Recall	F1-Score	Specificity
Negative	0.8102	0.7876	0.7939	0.9765
Neutral	0.9400	0.9416	0.9404	0.9200
Positive	0.9064	0.9142	0.9099	0.9559
Accuracy	0.9143			
Precision	0.9137			
Recall	0.9143			
F1-Score	0.9130			

**Table 3 bioengineering-11-00521-t003:** GPT-2 results.

GPT-2	Results			
	Precision	Recall	F1-Score	Specificity
Negative	0.5521	0.2821	0.3731	0.9654
Neutral	0.8107	0.9016	0.8541	0.7559
Positive	0.8002	0.8238	0.8103	0.8952
Accuracy	0.7872			
Precision	0.7024			
Recall	0.6513			
F1-Score	0.6610			

**Table 4 bioengineering-11-00521-t004:** XLNet results.

XLNet	Results			
	Precision	Recall	F1-Score	Specificity
Negative	0.5182	0.7195	0.6058	0.6322
Neutral	0.7870	0.8638	0.8236	0.8384
Positive	0.7354	0.7565	0.7458	0.8466
Accuracy	0.7490			
Precision	0.6194			
Recall	0.6643			
F1-Score	0.8615			

## Data Availability

The datasets presented in this article are not readily available because the data are part of an ongoing study and the ethical protocol prohibits the publication of these data.

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
