# Peer review of "A Comparative Sentiment Analysis of Greek Clinical Conversations Using BERT, RoBERTa, GPT-2, and XLNet"

_bioengineering, 2024, doi:10.3390/bioengineering11060521_

Round 1

Reviewer 1 Report

Comments and Suggestions for Authors

The parameter values of each trained AI-tool should be given in the main text. Indeed, details of these tools are lacking.

Figure 2 should not be a figure; it should be an array of equations. Type them.

Are positive emotions necessarily good? Toxic positivity is a real issue.

Comments on the Quality of English Language

Minor point: "GPT-2 demonstrated..." This verb does not sound appropriate. In fact, the verb "demonstrate" is misused in some sentences.

Author Response

- The parameter values of each trained AI-tool should be given in the main text. Indeed, details of these tools are lacking.
Thank you for your comments and feedback. I will ensure these details are added to the main text in the revised manuscript. I will add in Material and methods section: 

The BERT-base-uncased model consists of 12 transformer layers. Each layer has 12 attention heads and 768 hidden units, totalling around 110 million parameters. The RoBERTa-base model has 125 million parameters overall and is composed of 12 transformer layers with 768 hidden units and 12 attention heads. The XLNet-base-cased model consists of 12 transformer layers and 12 attention heads with 768 hidden units resulting in 110 million parameters. The configuration of GPT-2 includes 12 layers, with 768 hidden units and 12 attention heads, summing up to about 117 million parameters.

- Figure 2 should not be a figure; it should be an array of equations. Type them.

Your suggestion to convert Figure 2 from a graphical to an equation-based format is well-taken. I will revise this to an array of equations as recommended.

- Are positive emotions necessarily good? Toxic positivity is a real issue.

Your point about toxic positivity is very valid. I will refine the discussion section to add : 

Our comparative study successfully recognized sentiments in clinical conversations focused on hematologic malignancies. Since positive feelings are crucial for patient satisfaction it is important to note that an excessive focus on positivity, known as toxic positivity, can occassionally be harmful. Patients with hematologic malignancies often encounter complex emotions and overmphasis on positive feelings may potentially overlook real emotions of fear, pain and uncertainty. Incorporating sentiment analysis tools that are sensitive to negative or mixed emotions could help clinicians identify underlying negative sentiments more effectively. Finally, while positive sentiment is a sign of effective communication it is important to ensure that it does not overshadow the full range of expressed emotions.

- "GPT-2 demonstrated..." This verb does not sound appropriate. In fact, the verb "demonstrate" is misused in some sentences.

Thank you for pointing out the concerns with the use of the verb "demonstrated." I will revise the text to replace "demonstrated" with verbs such as "showed" and "exhibited".

Reviewer 2 Report

Comments and Suggestions for Authors

Congratulations on the study. I look forward to new research on this matter.

Author Response

Thank you very much for your kind words and encouragement! I am glad to hear that you found the study interesting. We are indeed planning further research in this area and hope to continue contributing valuable insights.